

# The vertical aerosol type distribution above Israel – 2 years of lidar observations at the coastal city of Haifa

**Birgit Heese**[1], **Athena A. Floutsi** TS1 [1], **Holger Baars**[1], **Dietrich Althausen**[1], **Julian Hofer**[1], **Alina Herzog**[1], **Silke Mewes**[1], **Martin Radenz**[1], **and Yoav Y. Schechner**[2]

[1]Leibniz Institute for Tropospheric Research – TROPOS, Permoserstr. 15, 04318 Leipzig, Germany CE1
[2]Viterbi Faculty of Electrical and Computer Engineering, Technion, Haifa, Israel

**Correspondence:** Birgit Heese (heese@tropos.de)

**Abstract.** TS2 For the first time, vertically resolved long-term lidar measurements of the aerosol distribution were conducted in Haifa, Israel. The measurements were performed by a Polly$^{XT}$ multi–wavelength Raman and polarization lidar. The lidar was measuring continuously over a 2-year period from March 2017 to May 2019. The resulting data set is a series of manually evaluated lidar optical property profiles. To identify the aerosol types in the observed layers, a novel aerosol typing method that was developed at TROPOS is used. This method applies optimal estimation to a combination of the lidar-derived intensive aerosol properties to determine the statistically most-likely contribution per aerosol component in terms of relative volume. A case study that shows several elevated aerosol layers illustrates this method and shows, for example, that coarse dust particles are observed up to 5 km height over Israel. From the whole data set, the seasonal distribution of the observed aerosol components over Israel is derived. Throughout all seasons, coarse spherical particles like sea salt and hygroscopically grown continental aerosols were observed. These particles originate from continental Europe and were transported over the Mediterranean Sea. Sea-salt particles were observed frequently due to the coastal site of Haifa. The highest contributions of coarse spherical particles are present in summer, autumn, and winter. During spring, mostly coarse non-spherical particles that are attributed to desert dust were observed. This is consistent with the distinct dust season in spring in Israel. An automated time–height-resolved air mass source attribution method identifies the origin of the dust in the Saharan and the Arabian deserts. Fine-mode spherical particles also contribute significantly to the observed aerosol mixture during most seasons. These particles originate mainly from the industrial region at the bay of Haifa.

## 1   Introduction

Automated, continuous height-resolved measurements were performed by the multi-wavelength Raman and polarization lidar Polly$^{XT}$ of TROPOS for 2 years from March 2017 until May 2019 at the Technion in Haifa, Israel, in collaboration with the Viterby Faculty of Electrical and Computer Engineering. This is the first long-term observation of the vertical aerosol distribution in Israel. Over recent decades, measurements of the aerosol distribution over Israel were mainly performed by ground-based in situ measurements (Ganor et al., 1998; Koçak et al., 2004) or by using satellite and sun photometer measurements (Israelevich et al., 2003).

The aerosol types that were observed frequently are marine aerosols (i.e., sea salt), urban and industrial pollution, and mineral dust from the surrounding deserts. Biomass burning aerosols produced by bush fires can also play a role (Heese et al., 2017). The source for marine aerosols is, of course, the Mediterranean Sea due to the vicinity of the site to it. In general, the particles have a radius from 0.01 to 100 µm in the remote marine boundary layer, and the large particles consist mainly of sea salt (Groß et al., 2013), whereas particles with a radius less than 0.2 µm are composed of non-sea-salt sulfate. Because of high relative humidity above 80 %, the particles are solution droplets and have a spherical shape

in the marine boundary layer (Charlson and Heintzenberg, 1995). However, according to Haarig et al. (2017b) the spherical sea-salt particles become cubic-like when the relative humidity decreases below 45 %, and their optical properties will change accordingly. The source of urban and industrial aerosols is the metropolitan area of Haifa with numerous electrical and chemical industries and oil refineries in the city and its surrounding area (Ganor et al., 1998). This aerosol type is especially existent in the lower part of the troposphere (Charlson and Heintzenberg, 1995). The particles are emitted directly into the atmosphere (primary aerosols) or can be formed by chemical reactions in the atmosphere (secondary aerosols).

The two main sources for dust that are important for the eastern Mediterranean are the Sahara and the deserts on the Arabian Peninsula (i.e., Kubilay et al., 2000; Nisantzi et al., 2015). The dust transport is governed by seasonal variations that are caused by the typical synoptical systems of this region (Israelevich et al., 2003). A study based on long-term observation in Beer Sheba from 1967 to 2003 by Dayan et al. (2008) quantifies the seasonal distribution of dust observations and accumulated dust from visibility observations and in situ measurements. They found that dust events occur mostly during the period from December to May and could correlate the observed dust events with the pressure systems that appear frequently during winter and spring: the Cyprus low and the Sharav cyclone. Both contribute substantially to the transport of dust towards Israel. Other typical synoptic systems are the Red Sea trough, the high pressure over Israel, high pressure to the east of Israel, and the Persian trough. These weather patterns correlate less with the dust observation in Israel. Nevertheless, so far, no measurements of the vertical aerosol type distribution over Israel are available, and knowledge on this topic is scarce.

In this paper, we analyze vertically resolved aerosol measurements performed by lidar and present a new retrieval scheme developed at TROPOS (Floutsi et al., 2019) in order to identify the aerosol types observed above Haifa. This retrieval scheme applies the optimal estimation method (OEM) to a combination of lidar-derived intensive aerosol properties (i.e., concentration-independent), to determine the statistically most-likely contribution per aerosol component in terms of relative volume.

The paper is structured as follows. In the next section, the Polly$^{XT}$ instrument at the measurement site is presented and the applied methods for the data evaluation are introduced: the lidar optical property profiles, the novel aerosol typing scheme, and the air mass source attribution method. An application of these methods is shown for a case study of multiple aerosol layers of different aerosol types. In the following section, the results of the aerosol typing and air mass source attribution for the entire measurement period are presented. From these results, the seasonal variations of the vertically resolved aerosol distribution over Haifa is derived. Finally, a discussion and an outlook conclude the paper.

## 2   Instrumentation and methods

### 2.1   Polly$^{XT}$ lidar and optical property profiles

The multi-wavelength Raman and polarization lidar Polly$^{XT}$ of TROPOS was installed in Haifa in March 2017 on the rooftop platform on a Technion building (Fig. 1). The measurement site is located at 32.8° N, 35.0° E at a height of 230 m above sea level. Haifa is a coastal site but the measurement site is located on one of the many hills of the city (Fig. 2). The lidar was deployed there for more than 2 years until May 2019 as part of the PollyNET (Baars et al., 2016), a network of permanent and campaign-based Polly$^{XT}$ stations.

The first generation of the multi-wavelength Raman and polarization lidar Polly$^{XT}$ of TROPOS (Althausen et al., 2009) was upgraded in 2014 to a new generation Polly$^{XT}$ instrument, as described by Engelmann et al. (2016). It has now the following features. It emits laser pulses at the three Nd:YAG wavelengths (1064, 532, and 355 nm) and has in total 12 detection channels in two receivers: 8 in the far-range receiver and 4 in the near-range receiver. The far-range receiver consists of three elastic backscatter channels at 1064, 532, and 355 nm and two Raman channels at 387 and 607 nm. Two channels measure the cross-polarized signals at 355 and 532 nm, and one channel measures the vibrational–rotational Raman inelastic signal from water vapor at 407 nm. The near-range receiver extends the signals of the two elastic channels at 355 and 532 nm and the two Raman channels at 387 and 607 nm down to a distance of 120 m above the lidar. The vertical resolution of the measurements is 7.5 m, and the data are stored every 30 s. From these measurements, a set of 3+2+2+1 optical property profiles is derived to allow for the proper characterization of particles in the atmosphere: three particle backscatter coefficient profiles, two particle extinction coefficient profiles, two particle linear depolarization ratio profiles, and one water vapor profile (Baars et al., 2016). The quality of the data is ensured by several calibration routines that are required by the European Aerosol Research Lidar Network CE2 (EARLINET) standards in the frame of AC-TRIS (Freudenthaler, 2016; Belegante et al., 2018).

### 2.2   Aerosol typing and discrimination scheme

To determine the aerosol types observed above Haifa by the lidar, a novel retrieval scheme was used that is currently under development at TROPOS (Floutsi et al., 2019). This aerosol typing scheme applies the optimal estimation method (OEM) to a combination of lidar-derived intensive aerosol properties (i.e., concentration-independent), to determine the statistically most-likely contribution per aerosol component in terms of relative volume, weighted against the microphysical and optical properties of the different aerosol components that are predefined. To be more specific, the intensive properties that are used in the retrieval are the lidar ratio and the

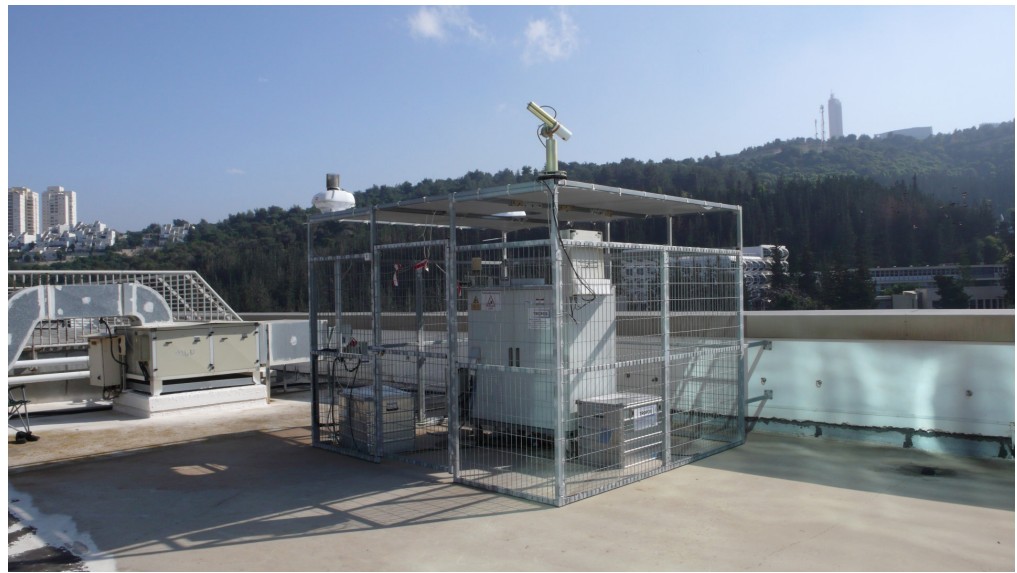

**Figure 1.** Polly[XT] on the rooftop platform at Technion, Haifa, in March 2017. The instrument is placed inside a protective cage to avoid unauthorized access. On top of the cage, a radar instrument is monitoring the sky for airplanes, and an AERONET sun photometer is measuring the aerosol optical thickness. In the background, the northern tips of the Carmel mountain range and the tower of the University of Haifa can be seen.

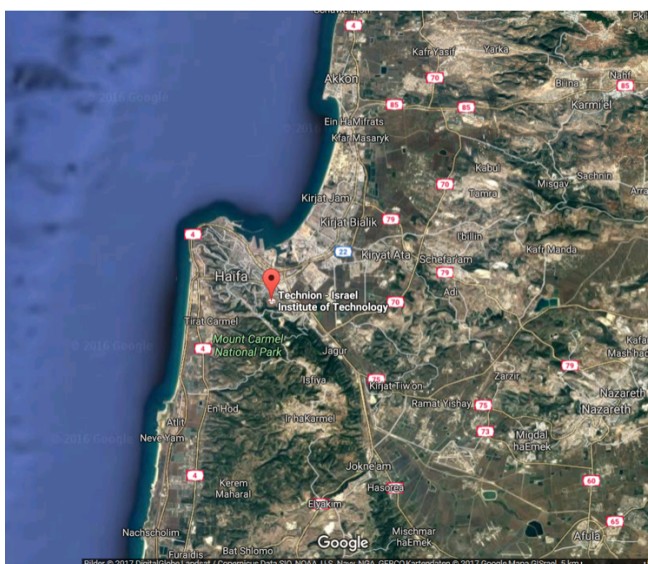

**Figure 2.** Map of Haifa: the location of the measurement site at the Technion is marked.

particle linear depolarization ratio at one or both wavelengths 355 and 532 nm.

Once a layer of interest has been identified and evaluated (in terms of optical properties), it is assumed that it consists of an external mixture of different aerosol components, occupying the total volume of the layer. In basic terms, OEM starts with an a priori state (assumed contribution per aerosol component – the aerosol components are described

**Table 1.** Pre-defined aerosol components with their optical properties at 355 nm/532 nm.

| Aerosol type | Lidar ratio (sr) | depolarization ratio (%) |
|---|---|---|
| Fine, spherical, absorbing (FSA) | 120/94 | 2/2 |
| Coarse, spherical (CS) | 17/19 | 3/3 |
| Fine, spherical, non-absorbing (FSNA) | 62/59 | 3/3 |
| Coarse, non-spherical (CNS) | 58/54 | 24/33 |

below), which is iteratively modified (using the Levenberg–Marquardt method; Rodgers, 2000) such that the modeled intensive optical properties match those observed by the lidar (within error range), resulting in the most probable estimated state (relative volume contribution per aerosol component). Then, this optimal solution undergoes a Pearson's chi-squared test to ensure that the optimal solution is statistically significant at 95 % confidence interval.

The aerosol components considered to contribute to an aerosol mixture in this OEM-based typing scheme are shown in Table 1. Two coarse-mode and two fine-mode aerosol components are considered: fine-mode, spherical, (strongly) absorbing (FSA) particles can be associated with aerosol originating from direct combustion processes. Coarse-mode, spherical (CS) particles can be sea-salt or ammonium sulfate aerosol particles. Fine-mode, spherical, non-absorbing (FSNA) particles are associated with typical pollution, which is less absorbing than FSA. FSNA may also be water-soluble (i.e., hygroscopic). Coarse-mode, non-spherical (CNS) particles are related to desert dust particles.

https://doi.org/10.5194/acp-22-1-2022 Atmos. Chem. Phys., 22, 1–16, 2022

The two coarse-mode and two fine-mode aerosol components have specific, pre-defined optical and microphysical properties. The optical properties are based on long-term CE3 Raman lidar observations. The microphysical properties (such as their shape, refractive index, and effective radius) and scattering properties of the four different aerosol components are consistent with the Hybrid End-to-End Aerosol Classification (HETEAC) scheme – the aerosol classification model for the EarthCARE space-borne lidar mission (Wandinger et al., 2016) – and are used directly in the OEM scheme as a priori information.

## 2.3 Air mass source attribution method

In order to identify the origin of the observed aerosols, we use the automated time–height-resolved air mass source attribution method TRACE that was recently developed by Radenz et al. (2021). This methodology can be applied using either ensemble backward trajectories or a Lagrangian particle dispersion model (LPDM) combined with a land cover classification scheme for a temporally and vertically resolved air mass source attribution. A simplified version of the MODIS land cover (Friedl et al., 2002) is used. The resulting land cover classes are water, savanna/shrubland, urban, barren, forest, grass/cropland, and snow/ice. In addition, custom defined regions that are relevant for the measurement location are used. For the site of Haifa, the regions Europe, the Sahara, the Arabian Peninsula, Far Eastern deserts, Persia CE4, and India were considered. Meteorological data are obtained from the Global Forecast System (GFS) analysis at a horizontal resolution of 1° (NCEP, 2000). In this study, we chose the LPDM approach using the most recent version of FLEXPART (Stohl et al., 2005; Pisso et al., 2019). A total of CE5 500 particles are used with their corresponding positions being stored every 3 h for 10 d back in time. A backward simulation is run every 3 h with height steps of 500 m for the period of interest. The reception height (i.e., the proximity of an air mass to the surface) is commonly chosen at 2 km, which is a widely applicable reception height threshold (Val Martin et al., 2018). However, in the following case study, a reception height of 5 km is used, since the observed aerosol layers extend up to 5 km. This height also takes into account that dust particles in particular are often lifted by small-scale processes in their source region and are not resolved by the meteorological data used for the transport simulation CE6 (e.g. Heese et al., 2009; Tesche et al., 2009b). The software package TRACE is available at Zenodo (Radenz, 2021).

## 3 Case study of multiple aerosol layers

In this chapter, the described methods are applied to a case study when multiple aerosol layers were observed above Haifa. Figure 3 shows the lidar backscatter signal at 1064 nm (a) and the volume depolarization ratio at 532 nm (b) during the night of 30 August 2018 at 18:00 UTC to 31 August 2018

at 06:00 UTC up to 6 km height above the measurement site. In these color plots, several aerosol layers can be identified already by eye. In the backscatter signal the lowermost layer is the planetary boundary layer (PBL). It is reaching up to 0.9 km height. The high backscatter signal at the upper boundary of the PBL is resulting from shallow clouds that become smaller CE7 after midnight. Above the PBL, a second layer is present that is reaching up to 2 km in the evening and is thinning out throughout the night. This layer is also visible in the volume depolarization ratio, as is the broad upper layer between 2.0 and 5.0 km. The volume depolarization ratio of the thicker, uppermost layer is much stronger, and the layer itself is slightly descending during the night. The visibility of these layers in the volume depolarization ratio already indicates that depolarizing particles are present here, in contrast to the PBL. The uppermost layer is slightly descending during the night. The identified layers are examined in more detail in a case study chosen in the cloud-free period from 01:20–02:44 UTC (red rectangle). During this period the second layer is extending between 0.9–1.6 km, and the upper layer between 2.1–5 km height.

## 3.1 Lidar optical properties

The first step of the lidar data analysis towards an aerosol typing is the determination of the optical property profiles. First, the particle backscatter coefficient (BSC) and the particle extinction coefficient (EXT) profiles were derived using the Raman method (Ansmann et al., 1992). From these, the ratio of the particle extinction coefficient to the particle backscatter coefficient, the so-called lidar ratio (LR), and the wavelength dependence of the backscatter and extinction coefficients expressed as the Ångström exponent, are calculated. Finally, the particle linear depolarization ratio (PLDR) is calculated using the polarization channels at 355 and 532 nm. The optical property profiles derived for the case study at the Haifa site are shown in Fig. 4.

### 3.1.1 Particle backscatter coefficient

The particle backscatter coefficient (BSC) profiles clearly show the boundaries of the observed layers. Inside the PBL, the BSC values are highest. The maximum values here are $5\,\mathrm{Mm}^{-1}\,\mathrm{sr}^{-1}$ at 355 nm, $2.8\,\mathrm{Mm}^{-1}\,\mathrm{sr}^{-1}$ at 532 nm and $1.5\,\mathrm{Mm}^{-1}\,\mathrm{sr}^{-1}$ at 1064 nm. The second layer can be identified by the maximum BSC in the lower part, around 1.25 km height. The values here are less than half of the BSC values in the PBL. In the broad, upper layer that is extending from about 2.1 to 5 km height, a substructure of three maxima at 2.8 km, at 3.7 km, and at 4.5 km is dominating the aerosol distribution on this day. The BSC values in these sub-layers are comparable to the ones in the second layer, except the uppermost sublayer, whose BSC values are slightly lower before they drop to zero above 5 km height. The effective resolution of the BSC profiles is 457.5 m.

Atmos. Chem. Phys., 22, 1–16, 2022
https://doi.org/10.5194/acp-22-1-2022

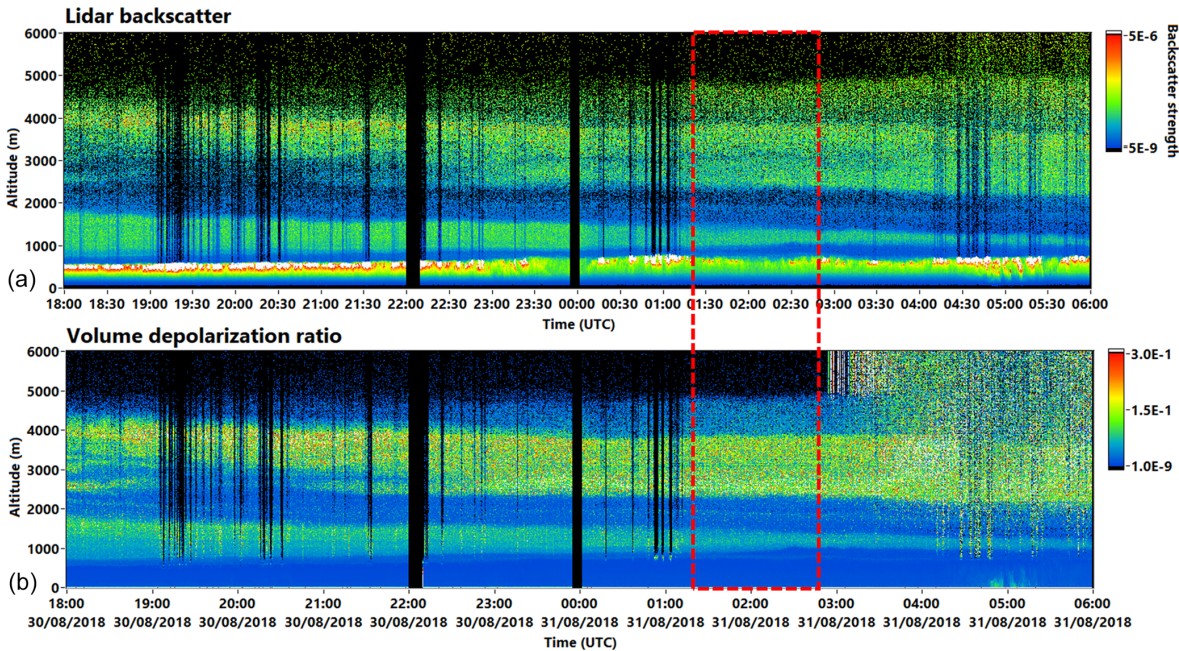

**Figure 3.** Color plot of the lidar backscatter signal at 1064 nm **(a)** and volume depolarization ratio **(b)** at 532 nm above Haifa on 30 August 2018 at 18:00 UTC to 31 August 2018 at 06:00 UTC. The time period used for the case study is marked by a red rectangle.

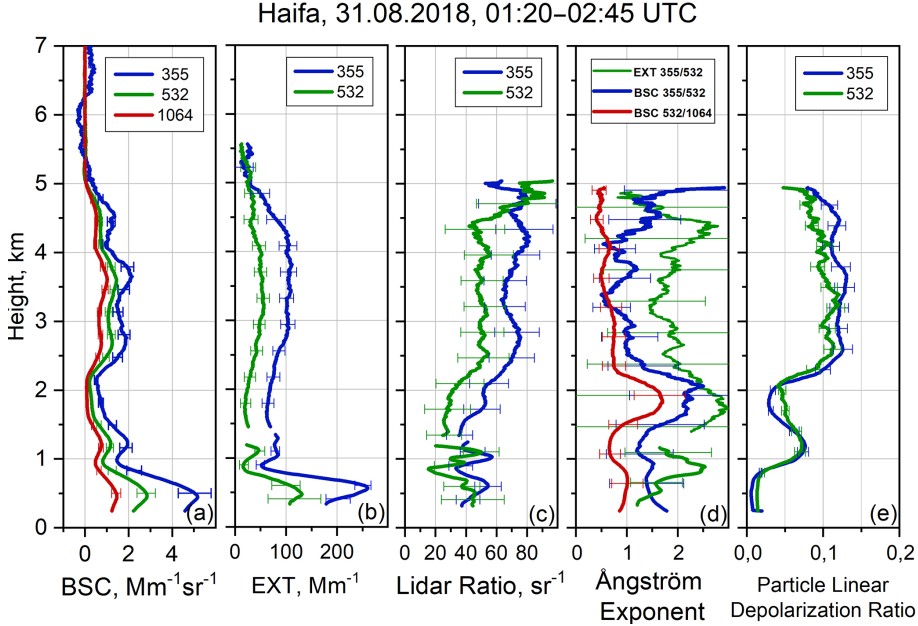

**Figure 4.** Profiles of the lidar optical properties particle backscatter coefficient (BSC), particle extinction coefficient (EXT), lidar ratio (LR), Ångström exponent, and particle linear depolarization ratio (PLDR) of the case study measured at Haifa on 31 August 2018. Near-range profiles are adapted for EXT, LR and the extinction-related Ångström exponent. The height range is starting above the measurement site.

### 3.1.2 Particle extinction coefficient

The particle extinction coefficient (EXT) has a much coarser effective resolution of 2000 m, because the backscattering efficiency from atmospheric nitrogen is 1000 times lower than the elastic backscatter efficiency. Therefore, the pro-

files shown in Fig. 4 show a smoother shape. The incomplete overlap between the laser and the receiving telescope in the far-range channels cause these profiles to start only at 1.3 km height. Therefore, below 1.3 km, the near-range profiles were used for the EXT and the derived properties

such as the LR and the extinction-related Ångström exponent. For the retrieval of the Raman BSC profiles, this overlap effect is eliminated due to its calculation by signal ratios (Wandinger and Ansmann, 2002; Engelmann et al., 2016). The height resolution used for the NR profiles was adapted to 2000 m by smoothing. Inside the PBL, the EXT values exceed $150 \, \text{Mm}^{-1}$ at 532 nm and $230 \, \text{Mm}^{-1}$ at 355 nm, indicating a strong absorbing aerosol component in this low layer. In the second layer, the EXT values decrease to much lower values before they increase again towards the upper layer. In the maximum of the broad, upper layer, between 3 and 4 km height, the EXT values reach $100 \, \text{Mm}^{-1}$ at 355 nm and about $50 \, \text{Mm}^{-1}$ at 532 nm. This is a factor of 2 and thus a strong wavelength dependence of the particle extinction coefficient.

### 3.1.3 Lidar ratio

For the profiles of the lidar ratio (LR), the same height resolution as the particle extinction coefficient (EXT) was adapted. It indicates whether the reflection or the extinction of light is predominant by the respective particles. Low LRs indicate more reflection of light and high LRs more scattering to other directions and absorption by the particles. The lidar ratio is widely used for aerosol typing, as described for example in Müller et al. (2007), Burton et al. (2012), Groß et al. (2015), Amiridis et al. (2015), and Baars et al. (2016). In the case of desert dust particles, the LR can even help to distinguish between dust from different deserts (Mamouri et al., 2013). In the case study, the LR in the PBL is around 50 sr at 355 nm and 40 sr at 532 nm. This indicates that urban pollution is likely to be the dominant aerosol type in the PBL. The second layer falls into the gap between far-range (FR) and near-range (NR) profiles. From the FR profiles, the LR in the upper part of the layer can be determined to around 40 sr at 355 nm and 30 sr at 532 nm. From the NR profiles it seems to be a bit higher in the lower part of the layer with values of 60 and 50 sr, respectively. Inside the broad, upper layer, the LR is again showing the structure of the three sublayers. At 355 nm, the maximum values are as high as 70 sr at 2.8 km and 80 sr at 4.2 and 4.8 km. At 532 nm the values are closer to 50 sr for all layers except the uppermost sublayer at 4.8 km. Here, the LR is in the same range as for 355 nm. These LRs indicate a different mixture of aerosol types in the individual layers.

### 3.1.4 Ångström exponent

The spectral dependence of the backscatter and extinction coefficients, the Ångström exponent, is dependent on the size of the observed particles. The values are low, 0.5 to 1, for coarse-mode particle size distributions, such as dust or sea-salt particles, and larger, with values above 1.5, for fine-mode particles from urban pollution or biomass burning aerosols (Hofer et al., 2020). The extinction-related Ångström exponent is more related to the absorbing abilities of the ob-

served particles. In the PBL, the values of the backscatter-related Ångström exponents are around 1 for the 532 to 1064 nm wavelength pair and around 1.5 between 355 and 532 nm. The extinction-related Ångström exponent is in the range between 1.5 and 2. These values indicate that rather small particles are present in the PBL. In the second layer, all three Ångström exponents decrease to 0.6 and 1.2 for the respective backscatter-related Ångström exponents and to 1.8 for the extinction-related Ångström exponent. These lower Ångström exponents indicate larger particles in this layer than in the PBL. Above this minimum, between 1.5 and 2.2 km, all three Ångström exponents clearly increase to values of 1.8, 2.3, and 2.9, respectively. These values indicate that here only small particles are present and mark a clear separation of the lower layer and the upper layer. In the upper layer, the Ångström exponents decrease again to 0.5, 0.9, and 1.5, respectively, so that also here larger particle are likely to be present. The extinction-related Ångström exponent is indeed still quite high, which may point to a non-negligible contribution of small, absorbing particles. In the uppermost sublayer, the backscatter-related Ångström exponent between 355 and 532 nm increases and the extinction-related Ångström exponent for the same wavelengths pair decreases to 1.5, in accordance with the equal LR. This low wavelength-dependence indicates that fewer large particles are present in this uppermost sublayer.

### 3.1.5 Particle linear depolarization ratio

Another important optical property that is measured by the lidar is the particle linear depolarization ratio (PLDR). It is a measure for the shape of the observed particles (Murayama et al., 1999). The profiles of the particle linear depolarization ratio show that both the second and the upper layer consist of non-spherical particles. In the PBL, the particle linear depolarization ratio at both wavelengths is practically zero. In the second layer, it rises to a maximum value of 0.07 at 532 nm and 0.07 at 355 nm at 1.25 km height. Then, it drops again towards the upper border of this layer. In the upper layer, the particle linear depolarization ratios are rising higher than 0.1 for the whole layer and have maxima of 0.13 at 355 nm and of 0.12 at 532 nm in the second sublayer from 3–4 km. These high depolarization ratios indicate that non-spherical particles with high reflection properties are present (Tesche et al., 2009a; Haarig et al., 2017a). The profiles of the particle linear depolarization ratio also shows the structure of the three sublayers inside the upper layer.

Using these optical property profiles, we can conclude so far that, due to high Ångström exponents and low depolarization ratios, small, spherical particles are likely to be present in the PBL. Towards the second layer, the Ångström exponent is decreasing, which suggests that larger particles are present. They also have higher depolarization ratios, but these are not as high as in the upper layer. Therefore, we would expect larger particles with a mixture of spherical

and non-spherical shapes. In the broad, upper layer, the low Ångström exponents and the higher depolarization ratios indicate the presence of large, non-spherical particles, like desert dust, that are mixed with smaller and, due to the higher extinction-related Ångström exponent, absorbing particles from pollution or other burning processes. As stated above, in the uppermost sublayer, the equal LR and backscatter-related Ångström exponent between 355 and 532 nm indicate that less large particles are present here. To further determine the type and origin of the particles in the observed layers, we applied the described OEM-based typing scheme and the air mass source attribution method.

## 3.2 Aerosol typing for multiple layer case study

We applied the OEM-based typing scheme to the detected aerosol layers of the case study of 31 August 2018. The results for the aerosol components in each aerosol layer are listed in Table 2. They are expressed in percentages, and the uncertainty range results directly from the retrieval's covariance matrix. It should be noted that while from a mathematical point of view relative volume contributions below 0 % (or above 100 %) are feasible, this is not the case from a physical point of view. Therefore, the errors hold only until the retrieved relative volume contribution of the aerosol component reaches 0 % or 100 %.

According to the scheme, the dominant aerosol types in the PBL with 86 % are FSNA particles. This aerosol type is associated with anthropogenic pollution related to the urban background and the industrial areas especially at the harbor bay of Haifa. The three other aerosol types represent only minor contributions.

In the lower layer, the main contribution to the aerosol composition is the coarse mode (CS) with 71 %. These particles can be sea salt as well as other aerosol types which have undergone hygroscopic growth. The FSA account for a percentage of 12 % in the second layer. They can be associated with the combustion products from the industrial and urban sources in Haifa.

The upper, broad layer with three sublayers is dominated by CNS particles, with a percentage of 74 %. Thus, this layer likely consists mainly of dust particles from the deserts. But FSNA particles and CS particles also account for a significant part of the aerosol components in this layer. This indicates that particles related to anthropogenic pollution and marine aerosols are also mixed into the elevated dust layer. FSA particles are negligible.

These results agree quite well with the findings when interpreting all available optical properties as discussed in Sect. 3.

## 3.3 Air mass sources of multiple layer case study

To identify the origin of the observed particles, the source identification method TRACE (see Sect. 2.3) was applied to the case study. The resulting air mass sources that were iden-

tified for the entire time period of Fig. 3 are shown in Figs. 5 and 6. For the particular case study from 01:20–02:45 UTC, the backward simulations at 00:00 and 03:00 UTC are relevant. The two major contributions to the air mass sources related to land cover classes are air masses that were transported over water and barren soil. Air masses that were transported over water are dominant in the lower layers. In the upper layers, the air mass that came from barren areas is strongly increasing. Small contributions come also from transport over forest, savanna/shrubland, and grass/cropland. The vertical distribution of the origin of the air masses related to the geographical region is shifting from Europe and the Sahara in the layers below 2 km to a decreasing fraction of the Sahara in favor of the Arabian Peninsula and Persian origin in the upper layers.

From these air mass source attributions, we can conclude that the high portion of coarse-mode, non-spherical particles identified by the aerosol typing in the upper layer originate from the deserts. The air masses were mainly transported over barren soil. The influences from the surrounding deserts and also from deserts further away are apparent. Above 4 km, the deserts on the Arabian Peninsula and small contributions originating from Persia are relevant. Below 4 km, the time spent over the Sahara is increasing and the time spent over the Arabian Peninsula is decreasing towards the lower part of the layer.

The coarse-mode particles that were identified in the second layer below 2 km mainly originate from the Sahara and the Arabian Peninsula, but also from Europe. This is directly linked to their residence time over barren soil. Another significant part of the residence time was spent over the Mediterranean Sea, so that sea-salt particles and particles that may have undergone hygroscopic growth are likely to be present. A small contribution to the residence time over forest, savanna/shrubland, and grass/cropland is increasing towards the ground.

Inside the PBL, these portions are highest. This reflects the influence of local sources in the lower layers. Here, mainly fine-mode, spherical, non-absorbing particles were identified by the OEM-based typing scheme. Although the residence time of the air masses over urban areas is rather short, the vicinity to the urban area is important inside the PBL and urban pollution is likely to be present. The long residence time over water in the lower layers is, of course, attributed to the vicinity of the Haifa site to the Mediterranean coast.

## 4 Results for the 2-year data set

The lidar data evaluation of the entire measurement period – from late March 2017 to mid-May 2019 – was performed manually. To achieve the complete set of lidar optical properties, only nighttime Raman measurements were used. The measurements were averaged over periods between 1 and 2 h during cloud-free and homogeneous atmospheric condi-

**Table 2.** Aerosol composition of the layers identified on 31 August 2018. TS3

|  | FSA | CS | FSNA | CNS |
|---|---|---|---|---|
| PBL ($< 900$ m): | $2 \pm 9\%$ | $8 \pm 20\%$ | $\mathbf{86 \pm 22\%}$ | $4 \pm 21\%$ |
| Lower layer (900–1600 m): | $12 \pm 13\%$ | $\mathbf{71 \pm 22\%}$ | $8 \pm 20\%$ | $9 \pm 19\%$ |
| Upper layer (2100–5000 m): | $1 \pm 12\%$ | $9 \pm 15\%$ | $16 \pm 17\%$ | $\mathbf{74 \pm 21\%}$ |

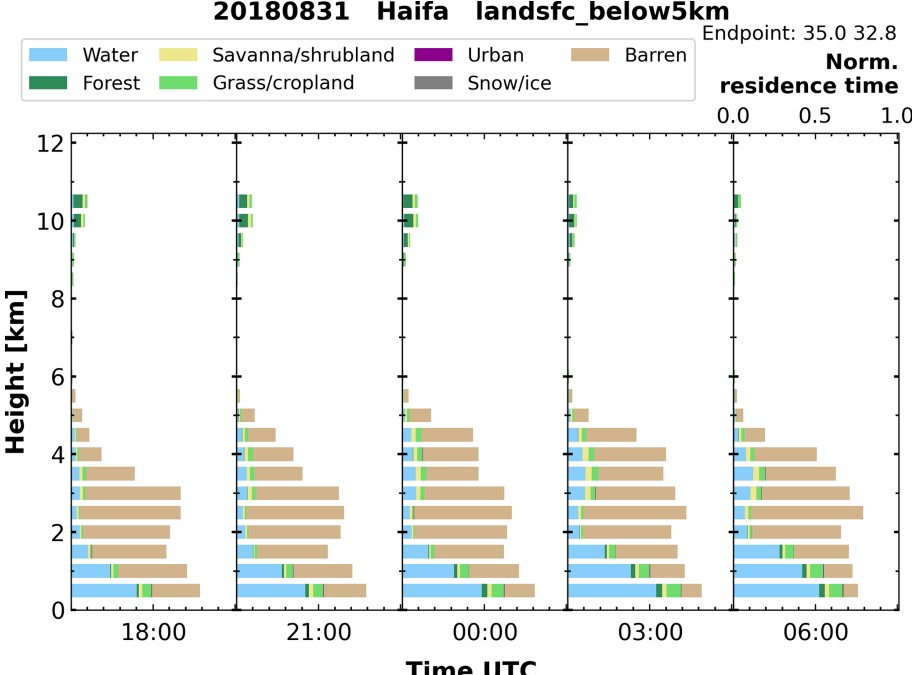

**Figure 5.** Air mass source estimate on 30 and 31 August 2018 for land cover classification based on FLEXPART particle positions. For each 3 h time step, the normalized residence time of the air mass over a particular land cover class is shown. The vertical step is 500 m and the reception height is 5 km.

tions. This procedure results in 397 evaluated lidar profiles in total. The optical properties of these profiles were derived carefully, and the existing aerosol layers in each profile were identified by visual inspection of the gradients. For each determined layer, the respective height interval and optical property values were compiled in a table for further analyses. Thus, in total, 1013 aerosol layers were identified. Occasionally, up to four layers – including the planetary boundary layer – were detected in one lidar profile. Out of these layers, only the ones that meet certain criteria were used as an input to the OEM-based typing scheme. More specifically, these criteria were the availability of lidar ratio and particle linear depolarization ratio at one wavelength at least (either 355 or 532 nm or both). From the 659 layers that were found to be suitable for evaluation by the OEM-based typing scheme, only 474 layers (71.9 %) lead to statistically significant retrievals and are therefore considered for further examination.

## 4.1 Aerosol typing – component predominance and mixture characterization

An overview of the vertical extent of the examined aerosol layers is shown in Fig. 7. It shows clearly that aerosol layers are present throughout the whole measurement period and that their vertical extent is varying with the seasons. During all seasons aerosol layers are confined in the lowest parts of the troposphere, on average between 740 m and 2 km. However, lofted layers at high altitudes were also observed, mostly between 2 and 4 km but also at much higher altitudes (up to 7 km). This was especially the case in spring and summer, but also during autumn and winter some lofted aerosol layers did occur. At this point, it should be mentioned that one exceptionally high layer that was found at an altitude between 11 and 12 km in spring 2018 is not shown in Fig. 7 (for visual clarity of the figure) but is considered in the analysis. The layers of Fig. 7 have been color-coded according to the aerosol component that is contributing to the mixture by at least 50 % in terms of relative volume contribution. This

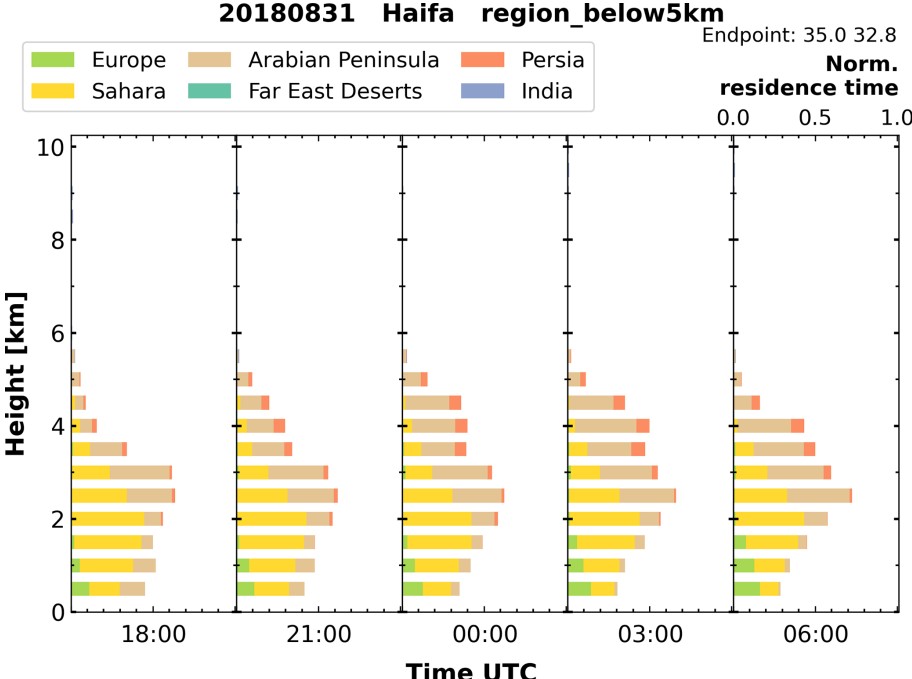

**Figure 6.** Same as Fig. 5 but for geographical areas.

rather large threshold is not able to reveal the layer composition, but it provides a first impression of the overall aerosol load composition. Most of the lower aerosol layers (below 2 km) have large contributions of CS particles. CS particles can be attributed mainly to marine particles, especially after taking into account the location of Haifa at the coast as well as the altitude of the layers inside the marine boundary layer. Above 2 km, most of the lofted layers are composed by FSNA and CNS aerosols. These aerosol components can be attributed to particles of anthropogenic origin and desert dust, respectively. Given the altitude of these layers, we can assume that these aerosol types were both subject to long-range transport.

The total composition of all the aerosol layers (not shown in Fig. 7) reveals that more than half of the layers consist of aerosol mixtures with two or more contributing aerosol components (277 layers), while fewer cases have a clear aerosol component predominance (relative contribution >80 %). To be more specific, 83 layers (17.5 %) were found to be dominated by CS, 67 layers (14.1 %) were dominated by FSNA contribution, 46 layers (9.7 %) were dominated by CNS contribution, and only 1 layer was found to be dominated by FSA (0.2 %). Pure layers (aerosol component relative contribution > 95 %) were observed only 8 times in the measurement period and consisted of CS aerosols. Since all layers consist of at least two aerosol components, examining the two dominant aerosol components contributing to the mixture can reveal information about the dominant mixture per layer. The most commonly occurring mixture was the one containing

CS particles (in 229 layers, mixed with either FSNA, FSA, or CNS), followed by FSNA, CNS, and FSA mixtures (131, 90, and 24 layers, respectively).

## 4.2 Seasonal variability

In this section, the seasonal variability of the different aerosol components found in the aerosol layers above Haifa is examined. In total, 143, 122, 145, and 64 layers were analyzed for spring, summer, autumn, and winter months, respectively. To be more specific, for all seasons, the frequency of occurrence of the aerosol components in the layers is determined in terms of relative volume contribution. In that way, we can evaluate how many times a specific range of relative volume contribution occurs for each aerosol component and give an overview of the seasonal aerosol distribution. Figure 8 shows the frequency of occurrence of the different aerosol components during all seasons. The height of the bars indicates the number of aerosol layers in which the aerosol component was present. The relative volume contribution of each aerosol component is represented by the position of the bar on the $x$ axes. In order to further support the typing scheme and identify the aerosol sources, the temporally and vertically resolved air mass source attribution method TRACE was used, as described in Sect. 2.3 The height profiles of the air mass sources for the Haifa station are shown in Fig. 9 using a reception height threshold of 2 km. In the following, the aerosol distribution during the four seasons are discussed on the basis of these two figures.

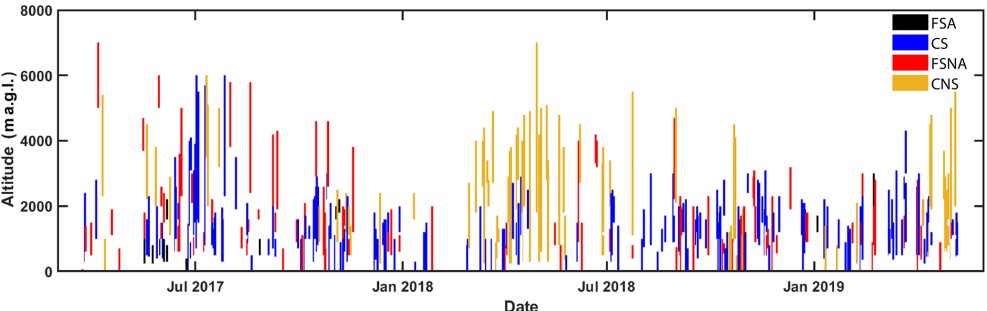

**Figure 7.** Overview of the altitude and extent of the analyzed aerosol layers above Haifa, Israel, between March 2017 and May 2019. Each layer is colored by its dominant aerosol component: black, blue, red, and yellow bars correspond to fine spherical, absorbing (FSA); coarse, spherical (CS); fine, spherical, non-absorbing (FSNA); and coarse, non-spherical (CNS), respectively.

### 4.2.1 Spring (March, April, May)

The frequency of occurrence of the different aerosol components during spring months March, April, and May 2017–2019 are shown in the first row of Fig. 8. The dominating aerosol component for the observed layers was the CNS component. A high occurrence of large contributions (relative volume above 60 %) is also observed for the CS component. The aerosol component least present (relative volume contribution below 20 %) was FSA. On average, aerosol mixtures are composed of $40 \pm 21$ % CNS, $31 \pm 19$ % CS, $21 \pm 18$ % FSNA, and $7 \pm 14$ % FSA.

The height profiles of the air mass sources for the Haifa station in spring are shown in the two upper left panels of Fig. 9. In the lowest 1 km, the most dominant air mass sources in terms of geographical area (a) were Europe and the Sahara, followed closely by the Arabian Peninsula. Air masses from Persia were less frequent and were observed up to 4.5 km. Above 1 km, the Sahara is the predominant source. In terms of land cover (b), air masses originating over water were the most frequent ones below 2 km and then again above 3.5 km. The second most frequent land cover was barren, reaching a peak between 2.5 and 3 km. Other land surfaces with significant air mass residence times were grass/cropland, savanna/shrubland, and forest. With the exception of water, the residence time of all other land categories decreases monotonically with height.

The predominant CNS particles can therefore be attributed to desert dust particles transported over the site mainly from the Sahara and the Arabian Peninsula. Air masses from Europe most likely carried FSNA and FSA particles, associated with aerosols of anthropogenic origin, urban background, and pollution. The latter two are expected to have local sources as well, given their low frequency of occurrence of high relative volume contributions. The overall high contribution of CS particles, usually associated with sea-salt particles, primarily composed of water-soluble, coarse sea-salt particles, can be explained by the fact that Haifa is a coastal city.

### 4.2.2 Summer (June, July, August)

The frequency of occurrence of the different aerosol components during summer months (June, July, and August 2017–2018) is shown in the second row of Fig. 8. The aerosol layers were dominated by CS particles, as indicated by the high occurrence of large relative volume contributions (greater than 70 %), followed by FSNA and CNS particles. The FSA component was the least present in the mixtures observed. On average, the observed aerosol mixtures consisted of $41 \pm 21$ % CS, $29 \pm 20$ % FSNA, $20 \pm 20$ % CNS, and $9 \pm 15$ % FSA.

The height profiles of the air mass sources for the Haifa station in summer are shown in the two upper right panels of Fig. 9. They reveal that in the lowest 4 km, the most dominant air mass sources in terms of geographical area (a) were Europe, and in terms of land cover (b) it was water. The Sahara and the Arabian Peninsula were also significant sources of air masses below 4 km, with the Sahara being the most prominent source between 6 and 9 km. Below 6 km, air mass sources also included Persia. Apart from water, dominant land surface categories included barren (peaking between 3.5 and 6 km), grass/cropland, savanna/shrubland, and forest. Land cover categorized as urban was only apparent below 3.5 km.

The air mass source attribution correlates well with the typing results. The dominance of the CS particles can be associated most likely with sea-salt particles picked up, while air masses were traveling towards Haifa crossing the Mediterranean sea. FSNA most likely originated from Europe, along with local contributions, while air masses originating from the Sahara and the Arabian Peninsula justify the relatively high contribution of CNS particles in the aerosol mixtures observed.

### 4.2.3 Autumn (September, October, November)

The third row of Fig. 8 depicts the frequency of occurrence of the different aerosol components during autumn months (September, October, and November 2017–2018). The aerosol component with the highest occurrence of large

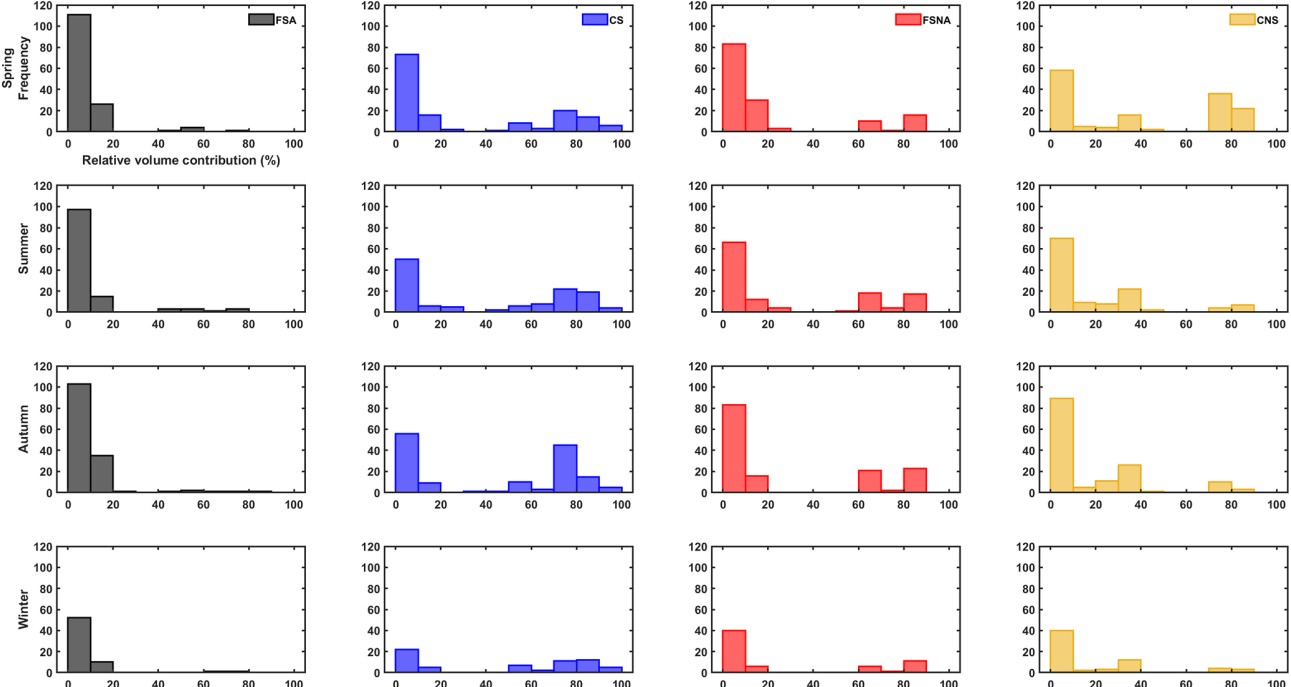

**Figure 8.** Frequency of occurrence (absolute) of the different aerosol components, in terms of relative volume contribution, during the four seasons. First row: spring months (March, April, and May 2017–2019); second row: summer months (June, July, and August 2017–2018); third row: autumn months (September, October, and November 2017–2018); and fourth row: winter months (December, January, and February 2017–2019).

relative volume contributions (greater than 80 %) was FSNA, followed closely by CS. Similarly to all previous seasons discussed, the FSA component was the least present in the mixtures observed. On average, the aerosol mixtures observed in the aerosol layers during autumn consisted of $44 \pm 21$ % CS, $28 \pm 20$ % FSNA, $19 \pm 20$ % CNS, and $8 \pm 15$ % FSA.

The most predominant air mass sources for autumn (Fig. 9, two lower left panels) were the Arabian Peninsula and Europe below 2 km, while above that altitude, the Sahara dominates in terms of geographical area (a). In terms of land surface (b), air masses spent significant time above areas characterized as water, barren, forest, savanna/shrubland, and grass/cropland.

The strong presence of FSNA can be correlated with the predominance of local and transported aerosols of anthropogenic origin from the Arabian Peninsula and Europe respectively. The CNS component of the aerosol mixture can be attributed to desert dust particles originating mainly from the Arabian Peninsula for altitudes below 2 km, and mainly from the Sahara at altitudes above 2 km.

### 4.2.4 Winter (December, January, February)

The last row of Fig. 8 shows the frequency of occurrence of the different aerosol components in terms of relative volume during winter months. The layers were mostly composed of CS particles, as indicated by their high occurrence of large

relative volume contributions (greater than 50 %), followed by FSNA particles. The aerosol component least present (relative volume contribution $< 20$ %) was FSA. On average, aerosol mixtures are composed of $47 \pm 21$ % CS, $26 \pm 20$ % FSNA, $19 \pm 20$ % CNS, and $7 \pm 15$ % FSA.

Profiles of air mass source for the aerosol layers observed above Haifa station in winter are shown in the two lower right panels of Fig. 9. In terms of geographical areas (a), below 2 km the most dominant air mass sources were Europe and the Sahara, followed closely by the Arabian Peninsula. Air masses from Persia were less frequent and only up to 2 km. Above 2 km and up to 12 km, the most predominant source is the Sahara. No air masses from the Far Eastern deserts and India fulfilled the 2 km reception height criterion. Most dominant land surface categories (b) were water, barren, and grass/cropland, with water having significantly higher residence time (especially above 4 km) compared to all the other land surface categories. The residence time of all other categories decreases monotonically with height.

Combining the aerosol information obtained from the OEM and the air mass sources from TRACE, we can conclude that the CS particles observed are most likely marine aerosols. The air masses traveling from the Sahara and Europe towards Haifa spent significant time above the water masses of the Mediterranean Sea allowing the uptake of sea-salt particles. The FSNA aerosol component can be attributed

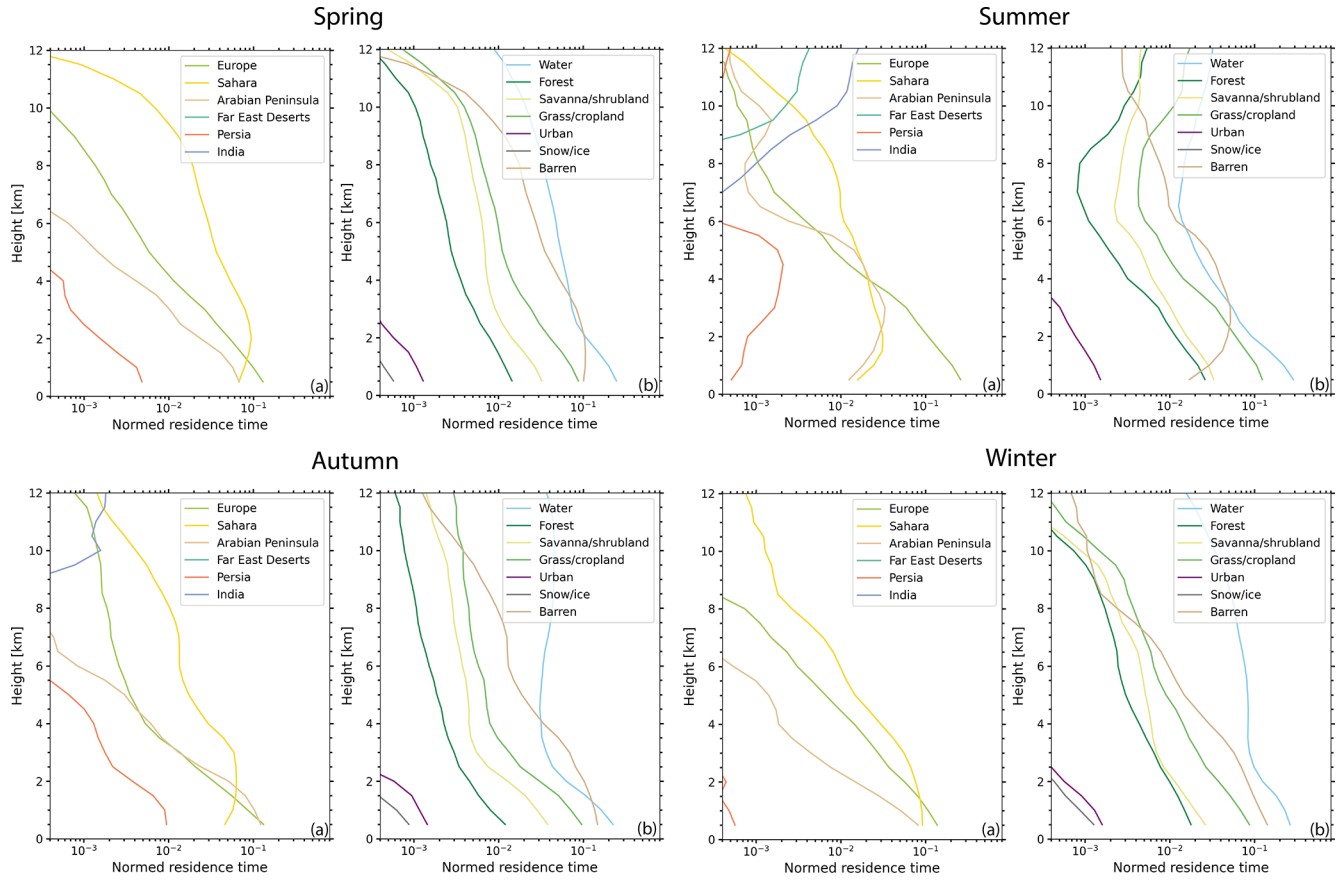

**Figure 9.** Air mass source estimate based on FLEXPART particle positions for the Haifa station for March, April, and May (spring season) 2017–2019 (upper left); for June, July, and August (summer season) 2017–2018 (upper right); for September, October, and November (autumn season) 2017–2018 (lower left); and for December, January, and February (winter season) 2017–2019 (lower right). The named geographical areas (a) and the land surface classification (b) are only for the periods with available lidar data (sub-sampled). The reception height threshold is 2 km.

to European and Arabian sources, and it is most likely composed of low or moderately absorbing aerosols, typically associated with aerosols of anthropogenic origin. The CNS particles observed are clearly desert dust particles, originating mainly from the Sahara region (long-range transport) but also from the Arabian Peninsula (locally produced or transported over short distances). Finally, the small contribution of FSA aerosols can be attributed to particles emitted from direct combustion processes, either produced regionally (Arabian Peninsula) or observed above Haifa after long-range transport events (e.g., Europe).

### 4.2.5 Overview

An overview of the statistics of the seasonal relative volume contribution per aerosol component is presented in Fig. 10. Overall, the component with the highest contribution (for all seasons except spring) is CS, followed by FSNA, CNS, and then FSA with the lowest contributions. The high contribution of CS to the aerosol mixtures was expected, given the

location of Haifa, the altitude of the examined layers, and the air mass source attribution. These CS particles can be correlated either with sea-salt particles or with other types of particles (e.g., continental aerosols) that, due to hygroscopic growth, grow to larger sizes and were therefore detected as large spherical particles to the aerosol mixture `CE8`. The contribution of FSNA can be explained partially by the fact that Haifa is surrounded by industries, including large petrochemical plants, an oil-fueled power station, a large cement factory, and petroleum refineries among other small industries and workshops (Ganor et al., 1998). FSNA aerosols not only have regional sources, but can be also long-range transported from Europe (sulfate-related particles) (Luria et al., 1996). During spring, the component with the highest contribution to the observed aerosol mixtures was CNS. This can be attributed to desert dust transfer by the prevailing synoptic patterns from the Sahara (Gkikas et al., 2016). Thermal Saharan lows developed south of the Atlas Mountains and, moving eastwards across the North African coast, induced south-southwesterly winds, thus favoring the transport of dust par-

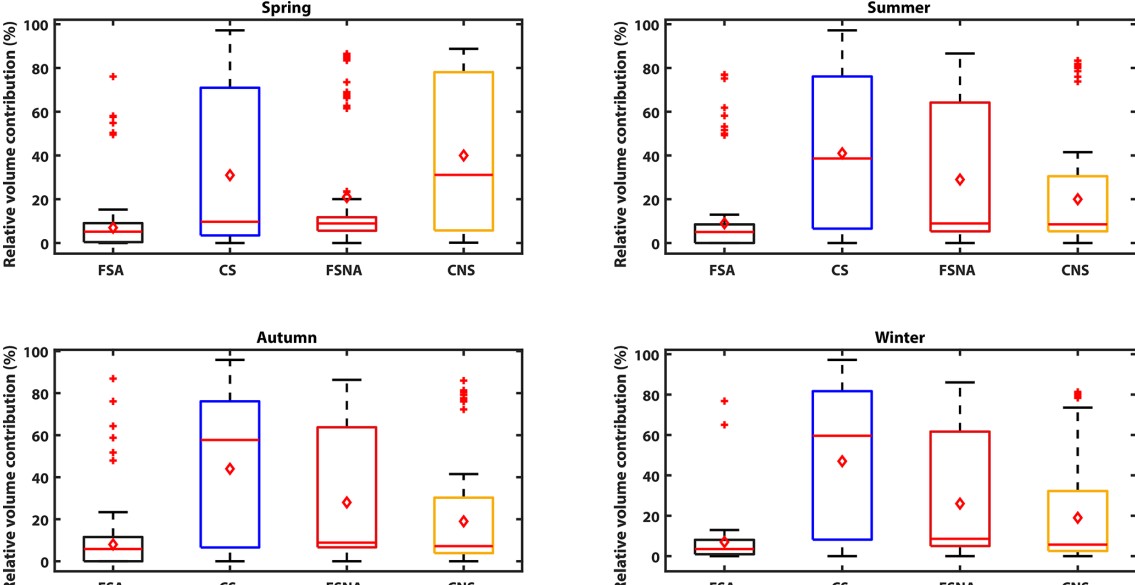

**Figure 10.** Seasonal statistics of the relative volume contribution of the different aerosol components. Minimum and maximum values are indicated by the lower and upper whisker, median and mean values by the red lines and rhombuses, respectively. The lower part of each box indicates the 25 % percentile and the upper part the 75 % percentile. Red crosses show the outliers.

ticles towards the Mediterranean and Europe (Moulin et al., 1997; Gkikas et al., 2014). The central and eastern parts of the Mediterranean Sea are commonly affected by dust transports under the aforementioned synoptic conditions as shown in previous studies (Barnaba and Gobbi, 2004; Papadimas et al., 2008; Gkikas et al., 2013; Floutsi et al., 2016).

## 5 Conclusions and outlook

For the first time, vertically resolved measurements of the aerosol distribution were taken at the coastal site of Haifa, Israel. The continuous long-term observations were performed with the multi-wavelength Raman and polarization lidar Polly$^{XT}$, which is part of EARLINET/ACTRIS and is considered to be one of the first long-term ACTRIS campaigns outside Europe. The measurements were conducted for 2 years and the resulting optical particle properties profiles were the basis for the characterization of the aerosol type distribution over Israel. A novel aerosol typing method developed at TROPOS was used for the distinct determination of the aerosol types. Based on this type separation, the vertical extent and the seasonal differences of the aerosol types were determined for the first time over this region in the eastern Mediterranean.

The analysis of the 2-year continuous data set revealed that a clear seasonal cycle in terms of the vertical aerosol distribution over Haifa exists. The highest extents of the aerosol layers were observed in the summer period with typical aerosol layer top heights up to 5–6 km. During wintertime, the aerosol layers are more tight to the ground with

typical aerosol top heights up to 2 km. Also, the occurrence of different aerosol types has a strong seasonal cycle. During the spring months (March–June), non-spherical particles have been observed frequently, as a result of intense and regular dust transport from the Sahara and the Arabian Peninsula towards Israel. The other months of the year (July–February) were dominated by coarse-spherical particles that originate either from marine sources or are a result of hygroscopic growth of fine-mode particles. Generally, the occurrence of pure aerosol types in this region of the world is sparse, and most of the time aerosol mixtures are observed. Fine-mode absorbing and non-absorbing particles are present all year round but contribute only partly to the overall aerosol load. The sources for this aerosol are anthropogenic either from the vicinity of Haifa or from long-range transport from outside Israel, or from both. Coarse-mode particles, either spherical or non-spherical, seem to be the dominant aerosol species above the local PBL.

Our results show for the first time a generalized picture of the type-separated aerosol distribution in the region of Haifa, Israel, and are a valuable basis for further studies on the aerosol radiative impact on climate and investigations towards aerosol–cloud interaction. In the frame of the collaboration with the Technion, the lidar measurements were also used as an anchor for a 3D wide-field sky scatterer tomography study. For a part of the duration of the lidar observations, extensive sky-view images were taken by all-sky cameras from multiple views, to observe and map clouds in 3D across time. The results of this study are described in Aides et al. (2020). A further study used the vertical lidar profiles at

Haifa to analyze the performance of the Multi-Angle Implementation of Atmospheric Correction algorithm (MAIAC) for cases of complex aerosol layering and mixing states in the eastern Mediterranean (Rogozovsky et al., 2021). Our results could also be used to improve parameterizations of aerosol transport models to enhance research on air quality in Israel. In the near future, further continuous aerosol lidars will be set up in the frame of ACTRIS within Europe but also in key regions outside Europe (e.g., in the dust belt). Currently efforts are being made to provide results from this type of measurements in near-real time so that such lidar observations may be in future assimilated in aerosol transport models and boost aerosol-related research in a new era. For that purpose, a new Polly$^{XT}$ lidar of the latest generation (Engelmann et al., 2016) was acquired by Tel Aviv University (TAU), Israel, and has been deployed at the rooftop of the Kaplun building at the TAU Campus since September 2019. Within this collaboration, the continuous observations of the aerosol distribution above Israel will be continued for the upcoming years.

**Data availability.** Lidar data are available upon request from the authors and data "quicklooks" are available on the PollyNET website (http://polly.tropos.de/, last access: 30 June 2021, Leibniz Institute for Tropospheric Research, 2021 TS4).

**Author contributions.** BH drafted and wrote the article. AAF developed the typing algorithm and wrote the respective parts of the article. She also conducted the TRACE air mass analysis, with assistance from MR. HB supervised her work. DA, JH, and BH implemented and maintained the instrument at Haifa. SM and AH analyzed the 2-year lidar data. YYS supported the lidar measurements and provided the measurement site. All authors contributed to the scientific discussion and reviewed and edited the paper.

**Competing interests.** The contact author has declared that neither they nor their co-authors have any competing interests.

**Acknowledgements.** We are grateful to Rotem Zamir, Vadim Holodovsky, Aviad Levis, and Amit Aides for assistance in the lidar operation in Haifa. This project is funded by the German-Israeli Foundation (GIF grant I-1262 401.10/2014) with support of the Norman and Helen Asher Fund. The work of Yoav Y. Schechner is conducted in the Ollendorff Minerva Center. Minerva is funded through the BMBF (German federal ministry for education and research). TROPOS has received funding from the European Union's Horizon 2020 research and innovation programme under grant agreement no. 654109.

**Financial support.** This research has been supported by the NAME OF FUNDER (grant no. GRANT AGREEMENT NO). TS5 The publication of this article was funded by the Open Access Fund of the Leibniz Association.

**Review statement.** This paper was edited by Anne Perring and reviewed by two anonymous referees.

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

## Remarks from the language copy-editor

CE1     Please note the slight edits to this affiliation (a space was added between the street and street number and the spelling of Tropospheric was corrected).

CE2     Please check.

CE3     Please check the word change.

CE4     Please note that Persia is an outdated term and as such is not allowed in our journals. Could this be changed to Iran throughout?

CE5     Please note that it is preferable that sentences do not begin with numerals.

CE6     Please check the sentence rewording.

CE7     Please check; the use of "less" here was grammatically incorrect.

CE8     Please check the sentence rewording.

## Remarks from the typesetter

TS1     Please check; Athena A. Floutsi or Athena Augusta Floutsi as in the MS records?

TS2     The composition of Figs. 3, 4, and 7 has been adjusted to our standards.

TS3     Please indicate in the caption why some values are denoted in bold.

TS4     Please add reference to reference list.

TS5     Please note that there is funding information given in the acknowledgements, but you did not indicate any funding upon manuscript registration. Therefore, we were not able to complete the financial support statement. Please provide the missing information and double-check your acknowledgements to see whether repeated information can be removed from the acknowledgements. Thanks.

TS6     Please ensure that any data sets and software codes used in this work are properly cited in the text and included in this reference list. Thereby, please keep our reference style in mind, including creators, titles, publisher/repository, persistent identifier, and publication year. Regarding the publisher/repository, please add "[data set]" or "[code]" to the entry (e.g. Zenodo [code]).

TS7     Please provide more information.

TS8     Please provide the edition, editors (if not authors), volume (if any), and a persistent identifier.

TS9     Please provide more information.

TS10     Please provide article number or page range.

TS11     Please provide more information.

TS12     Please provide article number or page range.

TS13     Please provide article number or page range.

TS14     Please check and confirm adjustements.

TS15     Please provide article number or page range.

TS16     Please check and confirm addition.

TS17     Please provide the edition, editors (if not authors), and a persistent identifier.

TS18     Please provide article number or page range.

TS19     Please provide article number or page range.