# Peer review of "The vertical aerosol type distribution above Israel - 2 years of lidar observations at the coastal city of Haifa"

_Atmospheric Chemistry and Physics, 2021_

## Author Comment (AC1)

We would like to thank the two reviewers for their helpful feedback and comments. We have responded to each general and specific Reviewer Comment point in detail. Our **answers** are printed in **bold face**. Our changes to figures in response to their suggestions are summarized below:

**In Fig. 4, the near range profiles are now included for the particle extinction coefficients and the Lidar ratio. The extinction-related Angstroem exponent profile was also added, including a near range profile. The title is correct for the year to 2018.**

**Fig. 5 and 6 have been replaced by similar ones, but for a shorter time period adapted to the same time period as for Fig. 3 (18:00 to 00:60 UTC).**

**Response to Review #1:**

General comments:

- [RC1]: A restructuring of the paper should be considered to first give a clear introduction to the general topic and the specific situation and needs in the eastern Mediterranean / Haifa region, followed by a description of the methodology (including all used methods used in this study e.g. the aerosol typing scheme), and then a clear presentation of the results.

**We rearranged the paper in the suggested way. It gives indeed a clearer overview over the used methods and separates the results for the case study and the two-year data set into own sections.**

- [RC1]: A comparison to the general aerosol situation from other measurements and to other lidar sites in the eastern Mediterranean is missing. This would in general enrich the content of this study to a more general overview.

**We would refrain from including a general overview over the aerosol situation from other lidar sites in the eastern Mediterranean in this paper. We would rather prospect this to an upcoming study including several ACTRIS stations in the region.**
**However, we included some references on the lidar measurements in Limassol to the introduction and discussion, that are dealing with lidar optical properties aerosol typing that add valuable information.**

*Mamouri, R.-E., Ansmann, A., Nisantzi, A., Kokkalis, P., Schwarz, A., and Hadjimitsis, D.: Low Arabian dust extinction-to-backscatter ratio, Geophys. Res. Lett., 40, https://doi.org/10.1002/grl.50898, 2013*

*Nisantzi, A., Mamouri, R. E., Ansmann, A., Schuster, G. L., and Hadjimitsis, D. G.: Middle East versus Saharan dust extinction-to-backscatter ratios, Atmos. Chem. Phys., 15, 7071–7084, https://doi.org/10.5194/acp-15-7071-2015, 2015*

- [RC1]: A height dependent consideration of the different aerosol types is missing. Do the pure or dominating aerosol types / aerosol situations mainly occur in the PBL or in the lofted layers?

**The height dependent aerosol distribution for the whole measurement period is illustrated in Fig. 7 and discussed in section 4.1. Also, the dominating aerosol types are presented here. In the following discussion on the seasonal variability the height distribution is also addressed. Fig. 10 gives a statistical overview over the height dependence of the aerosol types for all seasons.**

Specific comments:

- [RC1]: Figure 3 and 4: The different lidar optical properties could be discussed in a more integrated manner. The volume depolarization ratio is only mentioned for the second layer

**In the frame of the restructuring of the paper, we separated the description and the discussion of the lidar optical properties and added a paragraph integrating the different lidar optical properties and the resulting particles properties. Chapter 3.1.**

**Regarding the volume depolarization ratio, both layers are visible, but not the PBL. We rephrased this paragraph to clarify the role of the volume depolarization ratio:**

*Line 126: Above the PBL, a second layer is present that is reaching up to 2 km in the evening and is thinning out throughout the night. This layer is also visible in the volume depolarization ratio, as is the broad upper layer between 2.0 km and 5.0 km. The volume depolarization ratio of the thicker, uppermost layer is much stronger and the layer itself is slightly descending during the night. The visibility of these layers in the volume depolarization ratio already indicates that depolarizing particles are present here, in contrast to the PBL.*

- [RC1]: Figure 4: In the description of PollyXT lidar you are mentioning near range channels to perform measurements close to the lidar. Those measurement are missing in this analysis. Have they been used at all during this study? And how did you derive the results in the PBL?
- [RC1]: Figure 4: It would be very interesting to also include the Angstroem Exponent of extinction.

**The near range channels were used for all profiles when they were available. We include the near range profiles and the extinction related Angstroem exponent to Fig 4. and extended the discussion of the optical properties in the frame of the reconstruction of the manuscript. See Section 3.1.**

- [RC1]: What is the resolution of the lidar ratio? Did you use the same resolution for extinction coefficient and backscatter coefficient to derive it?

**Yes, it is the same resolution as the extinction, which is the limiting quantity in the calculation of the Lidar Ratio. We added the sentence:**
*The profiles of the lidar ratio (LR) have the same resolution as the particle extinction coefficient (EXT).*

- [RC1]: Case study: How do the OEM derived aerosol types (e.g. FSNA in the PBL) agree with the large contribution of water derived from the trajectory analysis? I guess from the trajectory analysis one would expect a larger contribution of marine aerosols. The different results should be better connected and discussed with one another.

**The air mass source attribution shows that the air masses spent significant time (residence time) above a surface characterized as water. That does not necessarily mean that sea salt particles were picked up from the air mass. In addition, a Fig. 6 indicates that for the same altitudes, the air masses that were above water partly originated from Europe, which is a source for FSNA aerosol (e.g., typical pollution).**

- [RC1]: Why do you only use 397 profiles of the 474 profiles with a statistically significant result? The distribution of these profiles over the year (i.e. to the different seasons) should already been mentioned at this point. Although, I recognized that it is mentioned later in the manuscript.

**This was maybe a bit confusing when describing lidar profiles and aerosol layers: 397 lidar *profiles* were evaluated from the whole measurement period. In the end, 474 aerosol *layers* were used for the aerosol typing. Due to the rearrangement of the manuscript, we now describe the data evaluation steps only once and in a more coherent way as an introduction to Section 4.**

- [RC1]: Figure 8: The volume contribution might be a bit misleading. Furthermore, for the large particles one would expect that they dominate the volume contribution, right? But that must not necessarily mean that they dominate the occurrence frequency. A better presentation or description of the results would be needed. Furthermore, I was wondering if a presentation of the relative frequency would not be clearer, as it would diminish the different sample frequency.

**In Fig. 8 (and throughout the whole manuscript) we are discussing about relative volume contributions of the different aerosol components and not about volume contributions. This is also clearly stated in the caption of Fig. 8.**

**The statement "for the large particles one would expect that they dominate the volume contribution" is not correct unless what is meant by the reviewer is 'volume concentration' instead of 'contribution' and that is true only if the coarse mode particles dominate the aerosol mixture (e.g., high relative volume contribution of coarse mode particles associated with high volume concentration of coarse particles). The presence of coarse mode particles does not necessarily mean that they dominate the relative volume contributions.**

- [RC1]: Figure 10: This information would give a good extension to the information shown in Figure 8 and the authors should consider to move this figure and give an extended description of the results there.

**We would rather leave the figure sequence as it is. Fig. 8 is the basis for the detailed discussion of the relative volume contributions of the different aerosol components during all seasons (see response above). We understood that Fig. 8 is maybe a bit difficult to interpret. Therefore, we moved the first mention of this figure to the introduction of the Chapter 4.2 (Seasonal variability) and explained how to read the height and position of the bars in the figure. Fig. 10 is representing a more statistical view on the seasonal differences and is therefore placed in the last subsection of this chapter. Both Figures are, of course, related to each other.**

- [RC1]: The content of the summary chapter is more like a discussion of the results. Maybe it should be considered to rename this section as 'discussion' and extend it, following the suggestions mentioned in the general comments.

**The last chapter is now called "conclusions and outlook" and gives a concluding summary of the results and an outlook how the vertically resolved aerosol observation in Israel will be continued.**

- [RC1]: The sentences are occasionally a bit long, e.g. in the Abstract.

**We shortened the long sentences to make them more readable.**

**Response to Review #2:**

General Comment:

- [RC2]: Some parts of the papers should be moved to another parts, in order to give a description of the all methods used in this study (I suggest to move to one section the explanation of lidar instruments, lidar optical properties profiles, the aerosol typing and discrimination scheme and air masses source methods) and perhaps you could include the case study as a subsection showing the application of this methods. And you should group the results in a section of results and discussion to give a clear presentation of the results and the discussion.

**We rearranged the manuscript in the suggested way. It gives indeed a clearer overview over the used methods and separates the results for the case study and the two-year data set into own sections.**

Specific comments:

- [RC2]: In the section regarding to the case study, you should discuss the figure 3 and 4 in a more comprehensive analysis. What happen with the PBL layer in the volume depolarization ratio figure?

**The PBL layer is not visible in the volume depolarization, since no depolarizing particles, like dust particles, are present. For clarity, we added the following sentence:**

*Line 129: The visibility of these layers in the volume depolarization ratio already indicates that depolarizing particles are present here, in contrast to the PBL.*

- [RC2]: In the same section, I suggest to number the optical properties profiles discussions (3.1.1 Particle BSC, 3.1.2 Particle EXT... etc..).

**In the frame of the rearrangement of the manuscript and the changes to Fig. 4, we extended the discussion of the lidar optical profiles (Section 3.1.) and numbered the subsections as suggested.**